# Behind the Curtain: In Silico and In Vitro Experiments Brought to Light New Insights into the Anticryptococcal Action of Synthetic Peptides

**DOI:** 10.3390/antibiotics12010153

**Published:** 2023-01-11

**Authors:** Tawanny K. B. Aguiar, Nilton A. S. Neto, Romério R. S. Silva, Cleverson D. T. Freitas, Felipe P. Mesquita, Luciana M. R. Alencar, Ralph Santos-Oliveira, Gustavo H. Goldman, Pedro F. N. Souza

**Affiliations:** 1Department of Biochemistry and Molecular Biology, Federal University of Ceará, Fortaleza 60451-970, CE, Brazil; 2Drug Research and Development Center, Department of Physiology and Pharmacology, Federal University of Ceará, Fortaleza 60430-275, CE, Brazil; 3Department of Physics, Laboratory of Biophysics and Nanosystems, Federal University of Maranhão, São Luís 65080-805, MA, Brazil; 4Laboratory of Nanoradiopharmaceuticals and Radiopharmacy, Zona Oeste State University, Rio de Janeiro 23070-200, RJ, Brazil; 5Brazilian Nuclear Energy Commission, Nuclear Engineering Institute, Rio de Janeiro 21941-906, RJ, Brazil; 6Faculty of Pharmaceutical Sciences of Ribeirão Preto, University of São Paulo, São Paulo 14040-903, SP, Brazil

**Keywords:** redox system, *Cryptococcus neoformans*, ROS metabolism, ergosterol, synthetic antimicrobial peptides

## Abstract

*Cryptococcus neoformans* is the pathogen responsible for cryptococcal pneumonia and meningitis, mainly affecting patients with suppressed immune systems. We have previously revealed the mechanism of anticryptococcal action of synthetic antimicrobial peptides (SAMPs). In this study, computational and experimental analyses provide new insights into the mechanisms of action of SAMPs. Computational analysis revealed that peptides interacted with the PHO36 membrane receptor of *C. neoformans*. Additionally, ROS (reactive oxygen species) overproduction, the enzymes of ROS metabolism, interference in the ergosterol biosynthesis pathway, and decoupling of cytochrome c mitochondrial membrane were evaluated. Three of four peptides were able to interact with the PHO36 receptor, altering its function and leading to ROS overproduction. SAMPs-treated *C. neoformans* cells showed a decrease in scavenger enzyme activity, supporting ROS accumulation. In the presence of ascorbic acid, an antioxidant agent, SAMPs did not induce ROS accumulation in *C. neoformans* cells. Interestingly, two SAMPs maintained inhibitory activity and membrane pore formation in *C. neoformans* cells by a ROS-independent mechanism. Yet, the ergosterol biosynthesis and lactate dehydrogenase activity were affected by SAMPs. In addition, we noticed decoupling of Cyt *c* from the mitochondria, which led to apoptosis events in the cryptococcal cells. The results presented herein suggest multiple mechanisms imposed by SAMPs against *C. neoformans* interfering in the development of resistance, thus revealing the potential of SAMPs in treating infections caused by *C. neoformans*.

## 1. Introduction

Currently, treatments against bacterial and fungal infections are limited due to the development of resistance to drugs by pathogens [1]. *C. neoformans* is a good example of a multidrug-resistant pathogen that causes dangerous infections worldwide [2]. Cryptococcosis and cryptococcal meningitis caused by *C. neoformans* mainly affect people with compromised immune systems. It is estimated 278,000 infections occur yearly in HIV-positive patients worldwide, leading to 181,000 deaths annually [3].

The high level of resistance presented by *C. neoformans* narrows down the number of drugs that can be used in treatments. For example, *C. neoformans* presents intrinsic resistance to caspofungin, which inhibits the enzyme (1→3)-β-D-glucan synthase and, nevertheless, perturbs the turnover of the fungal cell wall [1,3,4]. Therefore, combined treatment of amphotericin B (AmB) and flucytosine (FC) are commonly used to treat cryptococcosis infections [4]. However, prolonged exposure results in the emergence of cryptococcal populations resistant to this treatment, as well as to the toxicity of those drugs [5].

To cope with this problem imposed by *C. neoformans*, SAMPs have emerged as promising alternative molecules due to their mechanism of action, which is generally associated with membrane pore formation. This mechanism makes it difficult for microorganisms to acquire resistance, low toxicity, and allergenicity [6,7,8]. Recently, our research group reported the anti-cryptococcal potential of SAMPs PepGAT, PepKAA, *Rc*Alb-PepII, and *Rc*Alb-PepIII [7]. Studies on mechanisms of action revealed that SAMPs prompted membrane pore formation and apoptosis induced by DNA degradation in *C. neoformans* cells [7].

In this study, an in silico and in vitro approach provided new insight into the mechanism of action of SAMPs (PepGAT, PepKAA, *Rc*Alb-PepII, and *Rc*Alb-PepIII) against *C. neoformans*. In silico analysis revealed that three SAMPs bind to the PHO36 receptor of *C. neoformans,* inducing conformational alteration. In vitro analysis showed a high accumulation of ROSs in *C. neoformans* treated with SAMPs. In further experiments, it was determined that peptides cause a disbalance in redox enzymes and lactate dehydrogenase activity in *C. neoformans* cells. Additionally, SAMPs induced the decoupling of cytochrome *c* from the mitochondrion and inhibited ergosterol biosynthesis. Together, these findings strengthen the need for employment of these SAMPs against *C. neoformans* infections.

## 2. Results

### 2.1. ROS Accumulation in C. neoformans Cells

Recently, we showed that the SAMPs PepGAT, PepKAA, *Rc*Alb-PepII, and *Rc*Alb-PepIII presented an MIC_50_ against *C. neoformans* cells of 0.04, 0.04, 25, and 0.04 μg mL^−1^, respectively [7]. In the same study, some mechanisms of action were evaluated. In this study, new information about the mechanism of action is presented. All of the experiments were performed at MIC_50_ for all peptides. 

The first step analyzed whether the SAMPs were able to induce the accumulation of different types of ROS. The first analysis was conducted to evaluate the accumulation of anion superoxide (O2^•−^) (Figure 1). The experiment was designed using nitro blue tetrazolium (NBT), which is converted into formazan with a blue or cyan color in the presence of O2^•−^. As expected, the control cells of *C. neoformans* (Figure 1—DMSO panel) presented no blue or cyan dots, indicating no conversion of NBT in formazan and, thus, no accumulation of O2^•−^. In contrast, SAMPs-treated *C. neoformans* cells presented a blue or cyan color, suggesting the conversion of NBT by high levels of O2^•−^ into formazan (Figure 1: panel of peptides; blue or cyan dots—black arrow). Additionally, the quantification of formazan corroborated the data of light microscopy. All treatments presented the statistical significance of the control. 

In further experiments, the accumulation of H_2_O_2_ was induced by SAMPs in *C. neoformans* cells (Figure 2). The control cells treated with DMSO solution presented no accumulation of H_2_O_2_ (Figure 2). In contrast, all peptides induced ROS accumulation in *C. neoformans* cells. Based on the brightness fluorescence, RcAlb-PepIII, PepGAT, and PepKAA presented a higher accumulation of ROSs than RcAlb-PepII. Interestingly, in Figure 2, the light field shows that cells treated with PepGAT presented a conformational alteration, leading them to assume an elongated shape. This was not observed in the control cells. 

### 2.2. Synthetic Peptides Alter the Activity of Enzymes in ROS Metabolism

The detection of both O2^•−^ and H_2_O_2_ in *C. neoformans* cells treated with SAMPs led us to investigate the activity of the enzymes involved in redox metabolism. The first enzyme analyzed was the superoxide dismutase (SOD). As expected, control cells of *C. neoformans* presented the highest SOD activity (4.98 AU mgP^−1^). In contrast, *C. neoformans* cells treated with RcAlb-PepII and RcAlb-PepIII presented no SOD activity. Cells of *C. neoformans* treated with PepGAT and PepKAA still presented SOD activity, but the activity values were three and four times lower than those of the control cells (Figure 3A).

Regarding the catalase activity (CAT), the control cells presented the highest levels of activity compared to the treated cells (Figure 3B). As with SOD, RcAlb-PepIII did not present CAT activity (Figure 3B). In this case, no CAT activity was detected for PepKAA. RcAlb-PepII and PepGAT presented CAT activity levels three and five times lower than those of the control cells (Figure 3B).

For ascorbate peroxidase (APX), only cells treated with PepGAT presented no APX activity (Figure 3C). The cells treated with DMSO (control) presented the highest activity (3.43 AU mgP^−1^). In the case of the other SAMPs, RcAlb-PepIII, PepKAA, and RcAlb-PepII presented APX activity levels 7.4, 4, and 3.43 times lower, respectively, than *C. neoformans* cells treated with DMSO (Figure 3C).

### 2.3. Anticryptococcal Activity of Peptides Is Affected by Ascorbic Acid

To determine the role of ROSs (O2^•−^ and H_2_O_2_) in the activity of SAMPs against *C. neoformans*, the activity was observed in the presence of ascorbic acid (AsA, 10 mM) (Figure 4). As reported above, all of the experiments in this study were performed with MIC_50_ concentration. As shown in Figure 4A, in the absence of AsA, the SAMPs still presented MIC_50_ activity (Figure 4A white columns). However, in the presence of AsA, in which all ROSs (O2^•−^ and H_2_O_2_) were consumed, all SAMPs had affected activity levels. The most affected was RcAlb-PepII, which completely lost its activity (Figure 4A dashed columns). The other peptides still presented some activity, but the activity levels were below 20%. To prove the absence of ROS, a microscopic fluorescence analysis was conducted in the presence of AsA, which revealed that no ROSs were produced.

As Aguiar et al. [7] revealed, all SAMPs can induce pore formation. Herein, we aimed to evaluate whether this pore formation was ROS-dependent. The fluorescence microscopy of the propidium iodide uptake assay with AsA revealed that SAMPs did not induce pore formation. RcAlb-PepII entirely lost its activity in the presence of AsA (Figure 4A), and was unable to induce pore formation in *C. neoformans* cells (Figure 5). Likewise, PepGAT did not induce pore formation in *C. neoformans* cells in the presence of AsA (Figure 5). In contrast, RcAlb-PepIII and PepKAA still maintained some inhibitory activity and induced pore formation in *C. neoformans* cells in the presence of AsA, suggesting that this mechanism is not dependent on ROSs (Figure 5).

### 2.4. Synthetic Peptides Interfere in Other Metabolic Processes on C. neoformans Cells

Here, it was evaluated whether SAMPs could inhibit the biosynthesis of ergosterol in *C. neoformans* cells (Figure 6A). As expected, the control cells did not present any inhibition in ergosterol biosynthesis. In this assay, the control used for inhibition was itraconazole (ITR), inhibiting the biosynthesis of ergosterol at 47%. All tested SAMPs presented values of inhibition higher than those of ITR. RcAlb-PepII, RcAlb-PepIII, PepGAT, and PepKAA inhibited, respectively, 80%, 85%, 75%, and 89% of the biosynthesis of ergosterol in *C. neoformans* cells (Figure 6A). 

The energetic metabolism of *C. neoformans* was investigated after contact with SAMPs (Figure 6B,C). First, the ability of SAMPs to interfere with the activity of lactate dehydrogenase (LHD) in *C. neoformans* cells (Figure 6B) was analyzed. Control cells presented the highest activity of LDH (227.25 UA mgP^−1^) (Figure 6B). Apart from RcAlb-PepII (24.21 UA mgP^−1^), which presented LDH activity 10 times lower than the control cells, in the cells treated with RcAlb-PepIII, PepGAT, and PepKAA, no activity of LDH was detected (Figure 6B).

It was also analyzed whether peptides could induce the decoupling of Cyt c from the mitochondrial membranes of *C. neoformans* cells (Figure 6C). As expected, DSMO was unable to release Cyt c from the mitochondrial membranes of *C. neoformans*. In this experiment, the positive control that induced Cyt c from *C. neoformans* was H_2_O_2_, which presented the highest level of Cyt c decoupling of *C. neoformans* cells (Figure 6C). All SAMPs induced the decoupling of Cyt c from the mitochondrial membrane of *C. neoformans*. However, all of these values were below that of H_2_O_2_ (Figure 6C). Among SAMPs, the highest value for Cyt decoupling was presented by PepKAA. 

### 2.5. Computational Simulations

Aiming to produce more information about the mechanisms of action of SAMPs, we performed a docking analysis to try to explain more about the action of peptides. The protein chosen was the membrane receptor PHO36 from *C. neoformans*. First, the sequence of PHO36 from *Saccharomyces cerevisiae* was employed to fish the sequence of PHO36 from *C. neoformans*. After finding the protein sequence, the Swiss model server was employed to construct a three-dimensional (3D) model. Then, ClusPro 2.0 Web Server was used to perform the docking analysis. PHO36 is a transmembrane protein. Based on that, the peptides that did not interact in the transmembrane domain were only considered for docking analysis, as shown in Figure 7 (red dashed lines). Among the tested SAMPs, RcAlb-PepIII was the only peptide that interacted in the transmembrane domain; thus, the result was not considered. 

Contrary to RcAlb-PepIII, all the other peptides interacted with PHO36 in the extracellular domain (Figure 7). The binding energy of peptides with PHO36 was −632.98, −678.98, and 578.12 kCal mol^−1^, respectively, for RcAlb-PepIII, PepGAT, and PepKAA. An analysis of RMSD (root-mean-square deviation) indicated changes in the atomic position, and then in the 3D structure, of PHO36. The values of RMSD were 1.542, 0.876, and 1.247, respectively, for RcAlb-PepIII, PepGAT, and PepKAA. These values indicate that the interaction of peptides with PHO36 changed its structure and, thus, its functions in cells (Figure 7). The peptides interacted with the PHO36 receptor from *C. neoformans,* which was supported by hydrogen bonds and salt bridge-type interactions. 

## 3. Discussion

*C. neoformans* causes severe infections in immune-deficient patients, such as patients with transplanted organs and those in intensive care units [1,2,4]. As *C. neoformans* is resistant to several drugs used in its treatment, it becomes essential to search for bioactive molecules as an alternative to conventional treatment [9,10]. This study was developed based on this emergence to find new molecules in order to overcome the resistance of *C. neoformans* to drugs. Herein, we provide new mechanisms behind the activity of four synthetic peptides against *C. neoformans*. 

Our SAMPs demonstrated inhibitory activity (MIC50) in a previous study at low concentrations [7]. The mechanisms evaluated at that time were pore formation, DNA damage, apoptosis induction, and damage caused by peptides to the cell wall and pores in the membrane [7]. Based on previously published results regarding DNA damage and apoptosis induction, we began the analysis by evaluating the redox metabolism in *C. neoformans* cells after contact with peptides (Figure 1, Figure 2 and Figure 3). The induction of ROS overaccumulation in microorganisms by peptides was not a surprise, but it could explain how SAMPs act against *C. neoformans* [11,12,13,14]. 

In pathogenic fungi with controlled production, ROSs have many beneficial effects on pathogens, such as developmental process, increased virulence, biofilm formation, and infection [15]. On the other hand, ROSs are a byproduct of the natural metabolic process in cells. Without a proper scavenger system to balance their levels, ROSs could bring damage to cells by interaction with vital molecules such as DNA, lipids, and protein, leading to death [15]. 

Usually, H_2_O_2_ is the main molecule analyzed in experiments of ROS accumulation induced by peptides in cells because it is more stable and easy to evaluate [11,12,13]. Here, to better picture the redox state in *C. neoformans* cells, we analyzed the accumulation of •O_2_^−^ (Figure 1), which is one of the most unstable ROSs and is rapidly converted into H_2_O_2_ [15]. Our results revealed a higher accumulation of •O_2_^−^ in *C. neoformans* cells after treatment with peptides (Figure 1). Uncontrolled accumulation of •O_2_^−^ accelerates the oxidative damage to DNA molecules caused by iron. The •O_2_^−^ induces an increase in iron levels by releasing it from proteins and enzyme clusters. The free iron interacts with DNA molecules, oxidizing it and leading to fragmentation [16]. This result is in accordance with our previously published result that *C. neoformans* cells presented fragmented DNA after treatment with the same synthetic peptides [7]. To prevent the damage caused by •O_2_^−^, cells use the SOD enzyme to produce H_2_O_2_, which is more stable than •O_2_^−^, but still lethal [15]. Our results revealed a high accumulation of H_2_O_2_ in *C. neoformans* cells after incubation with peptides (Figure 2). 

Although H_2_O_2_ induces damage to DNA molecules, as does •O_2_^−^, it usually has other targets, such as proteins and lipids. In the case of lipids, H_2_O_2_ causes the oxidation of lipids in the membrane by a process known as lipid peroxidation. This process could lead to membrane destabilization and, consequently, pore formation, increasing membrane permeability [17,18]. In addition, H_2_O_2_ also interacts with proteins, damaging them and inhibiting their activity [19]. Recently, Branco et al. [19], using a proteomic approach, revealed that *Klebsiella pneumoniae* cells treated with a synthetic peptide presented a high accumulation of H_2_O_2_, followed by an increase in the accumulation of proteins involved in the recovery of proteins damaged by ROS. This result suggests that the higher levels of H_2_O_2_ are involved with protein damage, in agreement with our hypothesis. 

It is clear that synthetic peptides cause a perturbation in redox homeostasis of •O_2_^−^ and H_2_O_2_ (Figure 1 and Figure 2). However, more information about how peptides accomplish this is necessary. Based on this, the activity of scavenger enzymes was evaluated in *C. neoformans* cells. The enzymes evaluated were SOD, CAT, and APX (Figure 3). First, it is necessary to understand the role of these enzymes in ROS metabolism. SOD enzymes are involved in the conversion of •O_2_^−^ into H_2_O_2_; CAT and APX are responsible for converting H_2_O_2_ into H_2_O and O_2_ [15]. These enzymes are responsible for the delicate balance of ROS levels that distinguishes the beneficial from the harmful effects of ROSs. 

As revealed in Figure 3A, SAMPs-treated *C. neoformans* cells presented reduced SOD activity. This reduced SOD activity is responsible for two problems: (1) the reduced activity of the SOD enzyme is responsible for low levels of conversion of •O_2_^−^ into H_2_O_2_, leading to accumulation of •O_2_^−^ (Figure 1 blue or cyan dots). (2) Low activity of SOD in *C. neoformans* cells treated with peptides is still associated with H_2_O_2_, even if it is at a low concentration. However, the activity of CAT and APX (Figure 3B,C), which are involved in scavenging of H_2_O_2_, is also reduced in cells treated with peptides, leading to the accumulation of H_2_O_2_ in *C. neoformans* cells (Figure 2 green fluorescence). Therefore, synthetic peptides, by an unknown mechanism, insult the balance between SOD (converts •O_2_^−^ into H_2_O_2_), CAT, and APX (H_2_O_2_ in H_2_O and O_2_), producing a scenario wherein •O_2_^−^ and H_2_O_2_ (Figure 1 and Figure 2) accumulate at the same in *C. neoformans* cells, thus potentializing the damage caused by ROS. As far as we know, our study is the first to demonstrate ROS accumulation and propose how peptides induce it by negatively modulating the activity of redox enzymes involved in ROS metabolism.

There are few studies with similar results to ours regarding redox enzymes with peptides in yeasts, and even in *C. neoformans*. However, Neto et al. [20] reported that MoCBP_3_ from *Moringa oleifera* seeds also caused perturbation in the redox enzymes, leading to the accumulation of ROSs. In that case, the authors only measured the accumulation of H_2_O_2_. 

Our data revealed that ROSs are important to the anticryptococcal activity of SAMPs. However, one question arises: Is the antimicrobial action of peptides fully or partially dependent on ROSs? An experiment with the antioxidant AsA provided new clues for the answer to this question. In the presence of 10 mM of AsA, all peptides had affected activity (Figure 4A). The most affected peptide, RcAlb-PepII, completely lost its activity. Similar results were posted by Neto et al. [20] for an anticandidal protein that had its activity reduced by 60% in the presence of AsA. Fluorescence microscopy (Figure 4B–F) proved that there was no ROS accumulation in *C. neoformans* treated with peptides in the presence of AsA. 

A common mechanism of action of peptides against pathogens is the induction of pore formation on the membrane, leading to the loss of internal content and, subsequently, death [21,22]. The pore formation process depends on many aspects. It could be driven directly by the binding of peptides with lipids in the membrane or an indirect process driven by ROS species [13,15,18]. In a previous work, Aguiar et al. [7] showed that all synthetic peptides induced pore formation in *C. neoformans*. Here, as shown, the same peptides had activity in the absence of ROS, which was consumed by AsA. Therefore, we attempted to understand whether the ability of peptides to form pores is dependent on ROS accumulation. To do so, peptides were incubated with *C. neoformans* cells and AsA. After incubation, an iodide propidium uptake assay was performed. The result was quite surprising and exciting (Figure 5). The peptides RcAlb-PepII and PepGAT lost the ability to induce pore formation in *C. neoformans* membranes (Figure 5). For RcAlb-PepII, the result corroborates the loss of activity in the AsA (Figure 4A). 

The exciting results occurred with RcAlb-PepIII and PepKAA, which, even in AsA preventing ROS accumulation (Figure 4), induced pore formation in *C. neoformans* cells. This result suggests that the induction of pore formation by these peptides is ROS-dependent and might be driven by the direct interaction of peptides with the membrane. RcAlb-PepIII and PepKAA are cationic peptides with a net charge, respectively, of +1 and +3, and they have hydrophobic potential [8,23]. These features are important for pore formation in two ways: (1) positive charge is important to ionic interaction with the negative charge of lipid heads in the membrane, and (2) hydrophobic potential is critical for inserting peptides into the membrane’s hydrophobic core [8]. 

In our previous study [7], we observed that the presence of exogenous ergosterol affected the activity of peptides against *C. neoformans*, suggesting that peptides can bind to sterol in fungal membranes [7]. Therefore, we experimented with verifying whether peptides also inhibited ergosterol biosynthesis. In this experiment, the control was the antifungal drug ITR (Figure 6A). All peptides presented inhibition higher than ITR. ITR is an antifungal drug belonging to the azole class, whose main mechanism is to inhibit the ergosterol synthesis pathway. Our results demonstrate that peptides are more effective in inhibiting biosynthesis than ITR. Recently, the antifungal MoCBP_2_ protein, purified from *M. oleifera* seeds, could not inhibit the biosynthesis of ergosterol. New targets and different mechanisms in potential new drugs are important due to the resistance to the current antifungal [24].

All of our data suggest that synthetic peptides dysregulate the redox metabolism of *C. neoformans* cells. As we know, ROSs are natural byproducts of cell metabolism [15]. The energetic metabolism is essential to cell response to environmental insults because it provides energy, as NADPH and ATP are used to produce response proteins [25]. Even the regarding the importance of energetic metabolism to cells, studies reporting alterations caused by peptides in energetic metabolism are scarce. Herein, we attempted to understand whether peptides could cause perturbation in the production of energy by *C. neoformans*. First, the activity of the LDH enzyme in C. neoformans cells was analyzed after the treatment with peptides. All peptides dramatically reduced the activity of LDH (Figure 6B). 

LDH is involved in the carbohydrate metabolic pathway, and it catalyzes the conversion of pyruvate into lactate, regenerating the NAD^+^ from NADH [26]. This reaction is important to regenerate the NAD^+^ in order to maintain the glycolytic pathway, and to produce ATP and pyruvate in order to run the Krebs cycle [26]. Another experiment suggested that peptides interfere in the energetic metabolism of *C. neoformans* cells. The analysis of Cyt c decoupling from the mitochondrial membrane induced by peptides indicates that peptides interfere with mitochondria’s energy production. 

Inducing the decoupling of Cyt c from mitochondrial membrane peptides causes two problems for *C. neoformans* cells. First, Cyt c is a key molecule in the electron transport chain (ETC) to support ATP synthesis [27]. Inducing the decoupling of Cyt c peptides to destabilize the ETC leads to a depletion in the ATP levels of the cell. Second, the release of the mitochondrial membrane by Cyt c acts as a stimulus for cells to begin apoptosis. Thus, peptides may be inducing this event. It is essential to note that all peptides induced apoptosis in *C. neoformans* cells, as revealed by our previously published study [7].

In an attempt to find possible protein targets for peptides to induce these damages in *C. neoformans* cells, computational simulations were employed. The target chosen was a transmembrane protein known as PHO36. PHO36 is a receptor adiponectin-like protein involved in lipid and phosphate metabolism in yeasts [28]. PHO36 works with RAS proteins in the same pathway that is involved in several cellular events essential for the life of yeasts, such as division, apoptosis, longevity, differentiation, nitrogen, and carbon nutrition [28].

Herein, molecular modeling analysis revealed that RcAlb-PepII, PepGAT, and PepKAA interact with PHO36 in the extracellular domain, resulting in conformational alterations to its structure. By interacting with PHO36 and changing its structures, peptides inhibit PHO36 function in cells, negatively affecting several cellular processes in yeasts. Additionally, misfunction is related to a stimulus for apoptosis in yeast cells. Lopes et al. [29] recently reported that a synthetic peptide interacting with PHO36 from *C. albicans* induced ROS accumulation, DNA fragmentation, and apoptosis. Our results revealed that RcAlb-PepII, PepGAT, and PepKAA interact with PHO36 and cause the same damage. These results suggest PHO36 as a new target for antimicrobial activity mediated by synthetic peptides. 

## 4. Materials and Methods 

### 4.1. Fungal Strains, Chemicals, and Synthetic Peptides

*C. neoformans* (ATCC 32045) was obtained from the Department of Biochemistry and Molecular Biology at the Federal University of Ceará (UFC), Fortaleza, Brazil. The high-grade chemicals were obtained from Sigma Aldrich (São Paulo, SP, Brazil). The SAMPs PepGAT (GATIRAVNSR), PepKAA (KAANRIKYFQ), *Rc*Alb-PepII (AKLIPTIAL), and *Rc*Alb-PepIII (SLRGCC) were synthesized and purchased from the Chempeptide company (Shanghai, China). 

### 4.2. Antifungal Assay

The antifungal assay was performed following the methodology [7,30]. Yeasts were cultivated in YPD (yeast extract peptone dextrose) agar for fifteen days. After that, harvested in a YPD medium. Because the MIC50 found previously was 0.04 µg mL^−1^ [7] for all synthetic peptides, that was the concentration chosen at which to perform all studies of the mechanisms. Thus, 25 µL of YPD with cryptococcal cells (10^6^ cells mL^−1^) and 25 µL of SAMPs at their final concentrations (0.04 µg mL^−1^) were added and incubated for 24 h at 30 °C before each assay. The activity of SAMPs was also tested in the presence of 10 mM AsA to verify whether the activity of SAMPs was dependent of ROS overproduction [20].

### 4.3. Detection of ROS Overproduction

To evaluate the peptide-induced ROS generation (H_2_O_2_), a fluorometric assay with DCFH-DA (2′,7′ dichlorofluorescein diacetate) was performed. Briefly, after the antifungal assay, the samples were washed with NaCl 0.15 M and centrifuged (5000× *g* for 10 min at 4°C). Next, 9 μL of DCFH-DA was added, and cells were incubated for 20 min at 22 ± 2°C in the dark. Then, the samples were washed two times with NaCl 0.15 mM and centrifuged as described. Finally, cryptococcal cells were transferred to slides and observed with a fluorescence microscope (Olympus System BX 41, Tokyo, Japan) with an excitation wavelength of 535 nm and an emission wavelength of 617 nm [31].

Qualitative and quantitative assays for anion superoxide followed the example of Choi et al. [32]. For the qualitative assay, *C. neoformans* cells were treated with SAMPs. Then, they were washed with 0.15 M NaCl to remove the excess media. Afterward, cells were incubated with 0.1 mM of nitroblue tetrazolium (NBT) for 3 h at room temperature (22 ± 2°C) in the dark. Cells were then visualized using a light microscope (Olympus System BX 41, Tokyo, Japan). The quantitative assay was placed in the same way as the qualitative. The difference was that the quantitative assay was performed in 96-well plates, and the conversion of NBT to formazan was quantified at 630 nm in a microplate reader (Epoch, Biotek, Santa Clara, CA, United States). 

In addition, the same assay used to detect H_2_O_2_ was performed in the presence of 10 mM ascorbic acid (AsA) [20]. Moreover, the pore formation in the presence and absence of 10 mM AsA was assessed using the Propidium Iodide (PI) influx assay, following the methodology described in [7].

### 4.4. Redox System Enzyme Activity

#### 4.4.1. Catalase (CAT) 

The CAT activity was assessed according to [33] to evaluate the catalase activity. After the antifungal assay, conducted at the same conditions described previously, cells were washed three times with 0.15 M NaCl, resuspended in 0.05 M sodium acetate buffer pH 5.2, frozen for 24 h, sonicated for 30 min, and centrifuged for 10 min (10,000× *g* at 4 °C), and the supernatant was collected as described by [20]. A total of 200 μL of samples were incubated with 700 μL phosphate buffer with 50 mM potassium, pH 7.0, at 30 °C for 10 min. Subsequently, 100 μL of 112 mM H_2_O_2_ was added, starting the reaction. The mixture was placed into a quartz cuvette (1 cm^−1^) and absorbance was assessed. The reduction in absorbance at 240 nm was measured at intervals of 10 s until reaching 1 min. A decrease of 1.0 absorbance unit per minute was assumed to represent 1 unit of catalase activity (AU).

#### 4.4.2. Ascorbate Peroxidase (APX)

Ascorbate peroxidase activity was evaluated following the methodology previously described by Souza et al. [33]. After the antifungal assay, 800 μL tubes contained 50 mM potassium phosphate buffer, pH 6.0, which consisted of 0.5 mM of L-ascorbic acid and 100 μL of 2 mM hydrogen peroxide in 100 μL of either the treated sample or the control. Then, they were incubated at 30 °C for 10 min. The enzymatic activity was measured through ascorbate oxidation, indicating the action of the enzyme, for 1 min at 10 s intervals using the spectrophotometer at a length of wave of 290 nm. Ascorbate peroxidase activity was expressed (UA) by reducing absorbance by 0.01 at 290 nm, indicating the use of ascorbate to remove H_2_O_2_ by milligram of the protein (UA/mg).

#### 4.4.3. Superoxide Dismutase (SOD)

Superoxide dismutase activity was measured according to Souza et al. [33] in 96-well microplates. In triplicate, 1 M potassium phosphate buffer, pH 7.8 (10 μL), 1 mM 2,2′,2″,2‴- ethylenediaminetetraacetic acid (EDTA) (20 μL), 10 μL of Triton × 0.25%, 20 μL of 130 mM L-Methionine, 100 μL of samples in deionized water in the presence and absence of peptides (MIC50), and 100 mM of riboflavin (20 μL) were homogenized and kept in the dark for 5 min. Then, the reactional mixture was placed in a 96-well microplate, exposed to fluorescent light (32 W), and read at 630 nm in intervals of 1 min until reaching 5 min. All reagents without yeast extract (replaced by ultrapure water) were used as controls. The enzyme activity was measured as the difference between the absorbance recorded for the light reaction and the corresponding dark reaction (estimated per min). This was expressed in activity units (AU). One unit of SOD activity (1 AU) corresponded to the amount of the sample needed to inhibit the photoreduction of NBT by 50%.

### 4.5. Ergosterol Biosynthesis Inhibition 

The ergosterol biosynthesis inhibition was evaluated following the method described previously by Neto et al. [20]. Ergosterol content was calculated based on the following equations: % ergosterol + 24(28) [DHE = (Abs282/290) × F]/pellet weight (1)
% 23(28) DHE = [(Abs230/518) × F]/pellet weight(2)
% ergosterol = % ergosterol + 24(28) DHE − % 24(28) DHE(3)24(28) DHE refers to 24(28) dehydroergosterol, a class of sterol that presents an absorbance reading similar to that of ergosterol at 282 nm. F, in both equations, represents the factor for dilution in ethanol.

### 4.6. Lactate Dehydrogenase Activity 

The LDH Liquiform™ kit (Labtest Diagnóstica, BR) was used to evaluate lactate dehydrogenase activity, following the manufacturer’s instructions.

### 4.7. Cytochrome c Release

Because cytochrome *c* release is related to apoptotic events in cells, we evaluated the induction of cytochrome *c* release by peptides following the methodology described in Neto et al. [20]. The Cyt c was measured using a microtiter plate reader at 550 nm.

### 4.8. Bioinformatics Assays

#### 4.8.1. Molecular Modeling of PHO36 Receptor from the *C. neoformans* Genome

The *C. neoformans* amino acid sequence for PHO36 was taken using homolog genes from the NCBI database (http://www.ncbi.nlm.nih.gov (accessed on 10 November 2022)) with the BLAST tool, using the sequence of *Saccharomyces cerevisiae*. 

The 3D models of the PHO36 from *C. neoformans* were built by comparative modeling using the A chain of the revised crystals of the adiponectin receptors (PDB code: 5LXG and 5LWY) by means of the SWISS-MODEL (https://swissmodel.expasy.org/interactive (accessed on 10 November 2022)) [29]. All the checks and refinements in the models were performed following the protocol established by Lopes et al. [29]. The best 3D model was submitted to the simulation of interaction (receptor and each peptide).

#### 4.8.2. Molecular Docking

Molecular docking studies between the synthetic peptides (ligands) and the plasma membrane receptor of *C. neoformans* were performed using the protein–protein ClusPro 2.0 docking server (https://cluspro.bu.edu/login.php (accessed on 12 November 2022)), and the output files were analyzed using the PyMol program. 

### 4.9. Statistical Analysis

All experiments were performed three times, and the values are expressed as the mean ± standard error. GraphPad Prism 5.01 (GraphPad Software Company, Santa Clara, CA, USA) for Microsoft Windows was used to run the statistical analyses. All data obtained in the assays were compared using ANOVA followed by the Tukey test (*p* < 0.05).

## 5. Conclusions

The synthetic peptides evaluated in this study displayed anticryptococcal activity by multiple mechanisms of action. Synthetic peptides interfered with the redox enzymes, leading to the accumulation of ROSs, which are involved in cell death. It was also shown that some peptides induced pore formation in a ROS-dependent manner, while others did the same in a ROS-independent manner. All peptides caused perturbation in the energetic metabolism by inhibiting the activity of LDH and decoupling Cyt c from the mitochondrial membrane. Altogether, these results reinforce the potential of these synthetic peptides against *C. neoformans* and describe their activity along with a promise to develop new forms of treatment against *C. neoformans* infections. 

## Figures and Tables

**Figure 1 antibiotics-12-00153-f001:**
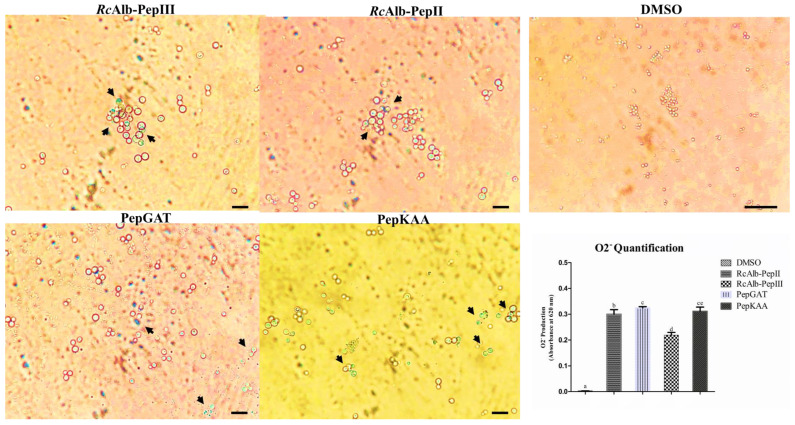
Qualitative and quantitative analysis of anion superoxide accumulation in *C. neoformans*. Light microscopy analysis of the conversion of NBT into formazan (blue or cyan dots–black arrows). The panel of DMSO represents the control cells, and other panels are treated *C. neoformans* cells with synthetic peptides. The inserted graphic represents the quantitative analysis of anion superoxide accumulation in *C. neoformans* cells. In control bar indicates 100 µm. In treated cells bar indicate 50 µm. The different lowercase letters indicate statistical significance at *p* > 0.05.

**Figure 2 antibiotics-12-00153-f002:**
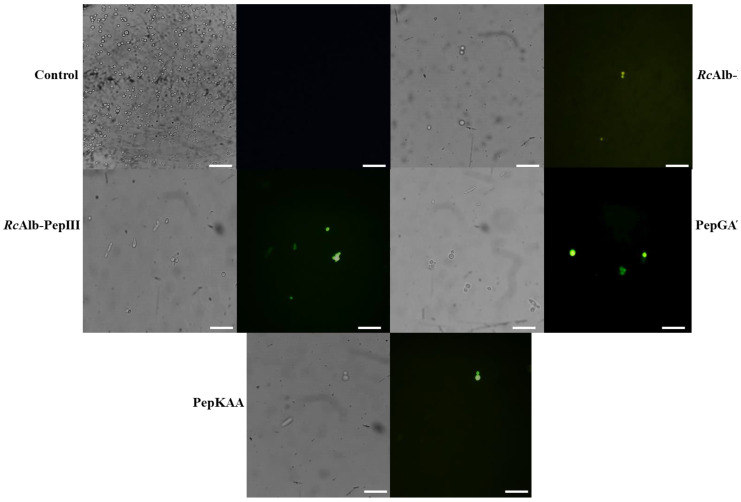
Hydrogen peroxide detection in *C. neoformans* cells. Green fluorescence revealed the overaccumulation of H_2_O_2_ in *C. neoformans* cells induced by synthetic peptides. Control cells were treated with 5% DMSO in 0.15 M NaCl. Bars indicate 100 μm.

**Figure 3 antibiotics-12-00153-f003:**
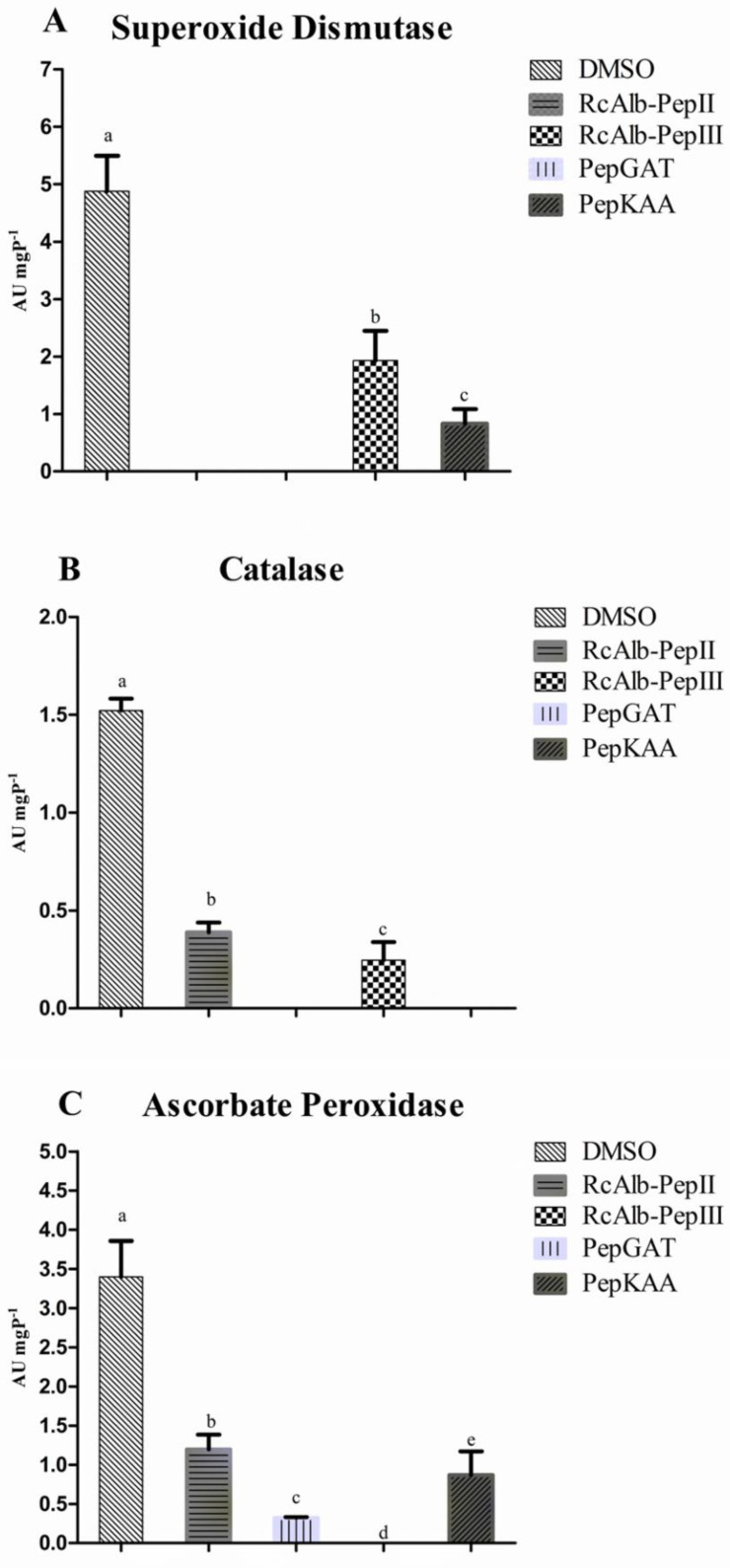
The activity of redox enzymes in *C. neoformans* cells. (**A**) SOD, (**B**) CAT, and (**C**) APX. All activities the enzymes were tested in *C. neoformans* cells treated and non-treated with synthetic peptides. SOD is an that enzyme that convert anion superoxide into hydrogen peroxide that is consumed by CAT and APX. The different lowercase letters indicate statistical significance at *p* > 0.05.

**Figure 4 antibiotics-12-00153-f004:**
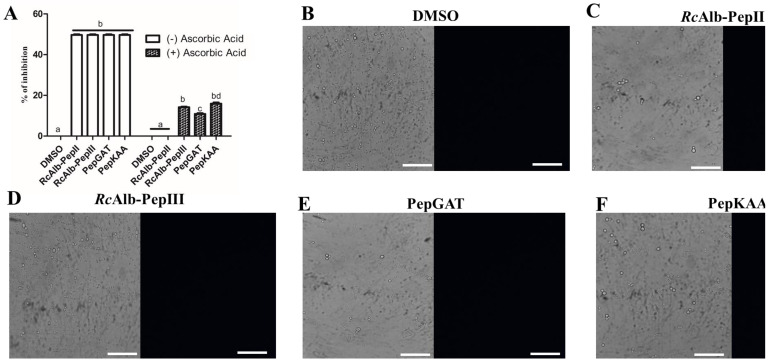
Effect of the antioxidant agent, ascorbic acid, in the activity of synthetic peptides against *C. neoformans*. (**A**) Inhibitory activity of synthetic peptides against *C. neoformans* in the presence of 10 mM of ascorbic acid. (**B**–**F**) Fluorescence microscopy analysis showed no detection of H_2_O_2_ in the cells of *C. neoformans* in the presence of 10 mM of ascorbic acid. Bars indicate 100 μm. The different lowercase letters in (**A**) indicates statistical significance at *p* > 0.05.

**Figure 5 antibiotics-12-00153-f005:**
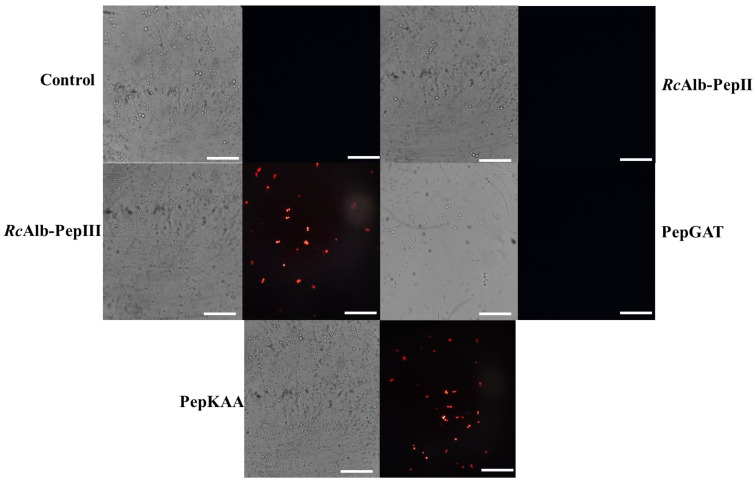
Effect of the antioxidant agent, ascorbic acid, in the membrane pore formation induced by synthetic peptides in *C. neoformans*. Propidium iodide uptake assay to evaluate the ability of synthetic peptides in induce pore formation in *C. neoformans* cells in the presence of 10 mM of ascorbic acid. Bars indicate 100 μm.

**Figure 6 antibiotics-12-00153-f006:**
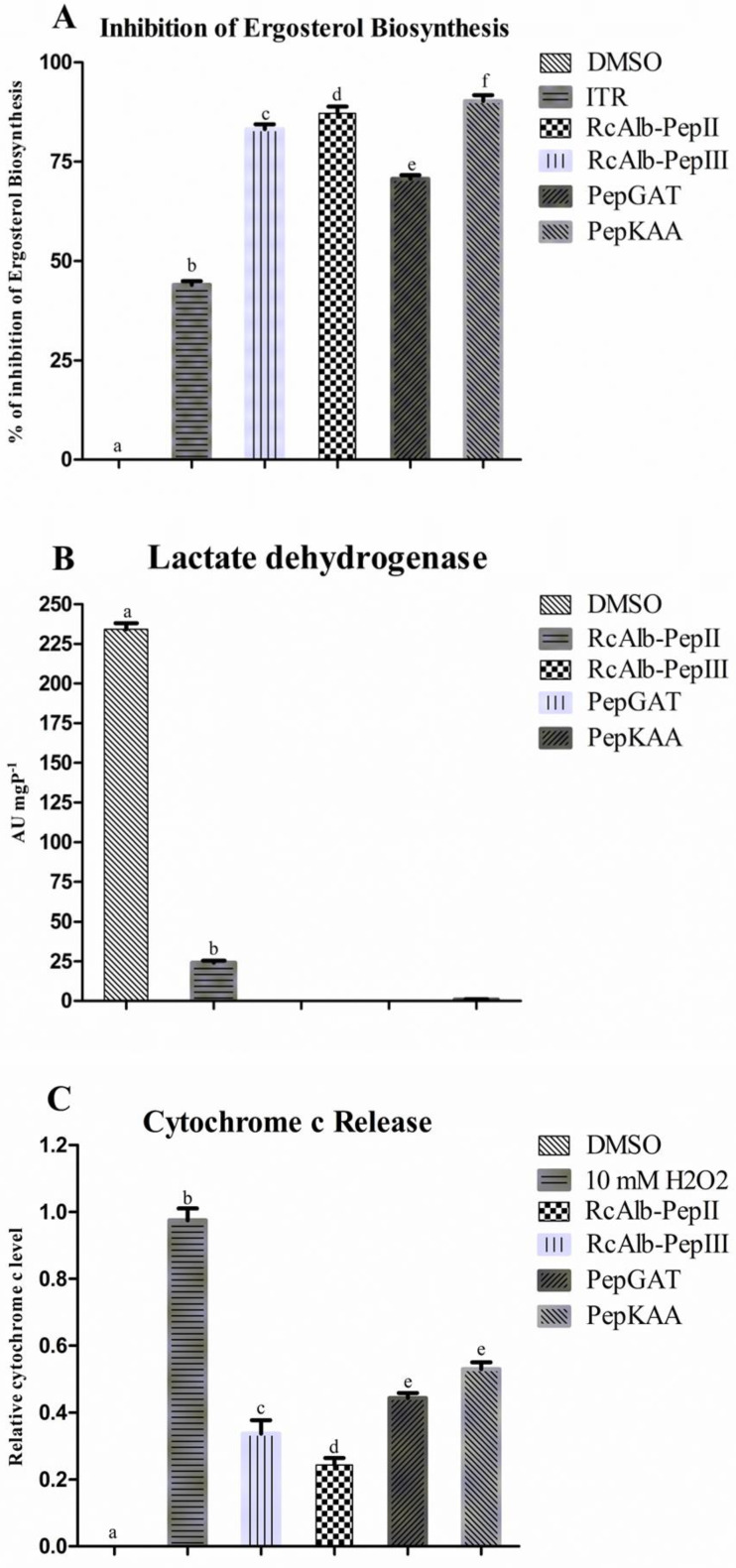
Effect of synthetic peptides in the cellular process of *C. neoformans*. (**A**) inhibition of the biosynthesis of ergosterol, (**B**) lactate dehydrogenase activity, and (**C**) release of Cytochrome c from the mitochondrial membrane. The different lowercase letters indicate statistical significance at *p* > 0.05.

**Figure 7 antibiotics-12-00153-f007:**
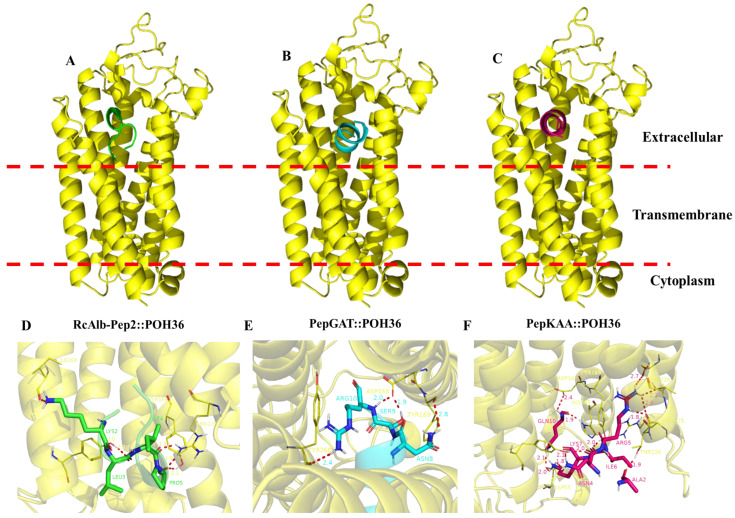
Molecular docking analysis of synthetic peptides and PHO36 receptor from *C. neoformans*. Overview of the interaction of peptides (**A**) RcAlb-PepII, (**B**) PepGAT, and (**C**) PepKAA with PHO36 from C. neoformans. Zoomed view of peptides (**D**) RcAlb-PepII, (**E**) PepGAT, and (**F**) PepKAA with PHO36 from *C. neoformans* showing amino acid residues involved in the interaction and distance.

## Data Availability

The data supporting this study’s findings are available upon request from the corresponding author.

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
