# Peer review of "Behind the Curtain: In Silico and In Vitro Experiments Brought to Light New Insights into the Anticryptococcal Action of Synthetic Peptides"

_antibiotics, 2023, doi:10.3390/antibiotics12010153_

Round 1

Reviewer 1 Report

The manuscript describes further investigations on the mechanism of action of synthetic peptides towards C. neoformans cells, previously described by the research group in 2021. The results and discussion seem solid and well structured. However, there are several English grammatical issues throughout the manuscript, which distracts from the scientific content. The entire document should be thoroughly revised for English grammatical. Overall, a very interesting and novel work.

Suggestions:

Line 22: neoformans not in italic.

Line 25: ROS (Reactive Oxygen Species).

Line 48: The number OF 278,000 infections…

Line 53: amphotericin b should not start with capital letter.

Line 131: “cells” should be in italic.

Line 135: 2.4. Synthetic peptides interfere in other metabolic processes on C. neoformans cells

Line 136: C. neoformans not in italic.

Line 139: itraconazole (lower case).

Line 185: C. neoformans not in italic.

Line 268: C. neoformans not in italic.

Line 268 and 270: The word “here” is used twice, please change one of them to avoid repetition.

Line 313: C. neoformans not in italic.

Line 356: YPD (yeast extract peptone dextrose) agar.

Line 393: H202 (numbers not subscripted).

Line 413 and 426: MIC50 and MIC50, inconsistency.

Line 493: P.F.N.S. repeated.

Please, the legend of Figure 3 could be extended and more detailed. What the letters on the top of each bar mean?

Figure 4 legend: Each sentence should start with capital letter.

Figure 5 legend: C. neoformans not in italic.

Author Response

Authors´ Response to Reviewer #1

Reviewers' comments:
Reviewer #1 - Comments to the Author

The manuscript describes further investigations on the mechanism of action of synthetic peptides towards C. neoformans cells, previously described by the research group in 2021. The results and discussion seem solid and well structured. However, there are several English grammatical issues throughout the manuscript, which distracts from the scientific content. The entire document should be thoroughly revised for English grammatical. Overall, a very interesting and novel work.

Authors’ Response

Dear Reviewer #1

We are thankful for you expending time to review our manuscript. Certainly, your suggestion will bring the manuscript to a higher scientific level.

The manuscript was revised to solve the problems you have noticed. Thank you.

Reviewer #1 Suggestions:

Line 22: neoformans not in italic.

Line 25: ROS (Reactive Oxygen Species).

Line 48: The number OF 278,000 infections…

Line 53: amphotericin b should not start with capital letter.

Line 131: “cells” should be in italic.

Line 135: 2.4. Synthetic peptides interfere in other metabolic processes on C. neoformans cells

Line 136: C. neoformans not in italic.

Line 139: itraconazole (lower case).

Line 185: C. neoformans not in italic.

Line 268: C. neoformans not in italic.

Line 268 and 270: The word “here” is used twice, please change one of them to avoid repetition.

Line 313: C. neoformans not in italic.

Line 356: YPD (yeast extract peptone dextrose) agar.

Line 393: H202 (numbers not subscripted).

Line 413 and 426: MIC50 and MIC50, inconsistency.

Line 493: P.F.N.S. repeated.

Please, the legend of Figure 3 could be extended and more detailed. What the letters on the top of each bar mean?

Figure 4 legend: Each sentence should start with capital letter.

Figure 5 legend: C. neoformans not in italic..

Authors’ Response

Thank you, all suggestions. All of them were fixed and accepted.

Reviewer 2 Report

Dear authors, there are several comments and questions to your article. This article (2) is a continuation of Antifungal Potential of Synthetic Peptides against Cryptococcus neoformans: Mechanism of Action Studies Reveal Synthetic Peptides Induce Membrane–Pore Formation, DNA degradation, and apoptosis (1). The experiments in both papers are well designed and deserve close attention. However, due to the endless references to article (1), article (2) lacks data. It is necessary to expand the Introduction section, more clearly outlining the range of issues addressed in the text.

It is necessary to characterize the studied peptides. Give the amino acid sequence and describe in detail what are the differences between the peptides, in relation to the possible mechanism of action. You should also understand the abbreviation for peptides. In the abstract, SP is used, in the introduction SAMP, hereinafter it is found simply - peptides. The choice of specific peptides should be justified: for what reasons this sequence was recognized as promising.

Most of the questions are caused by drawings.

Figure 1. No scale lines. Control and experiment were taken at different magnifications. The white balance is out of alignment, causing blue to appear green. The text indicates "blue dots", which are not on some pictures.

Figure 2. In the PepGat (fluorescence) figure, there is a green spot that does not correspond to any of the cells shown in the illustration without fluorescence. In other figures, elongated cells are noticeable, if this is an artifact, it must be indicated in the text.

Figure 4. What method was used to obtain the data for 4A? What does "percent inhibition" mean? It would be more informative to provide data on cell concentration after exposure to peptides in each experiment (with and without ascorbic acid). This question arises since in the above images the number of cells per area unit under the influence of peptides is noticeably lower than in the control. At the same time, the text of the article says that the peptides do not exhibit antifungal activity. In the signature there is no decoding of the letters «a, b, c, d» given on 4A.

Figure 5. The same. The concentration of cells under the influence of peptides is lower than in the control, in the absence of fluorescence. It is necessary either to replace the pictures, or to explain this effect.

In all signatures the indicate must be replaced with a correspond to.

MIC or MIC50? Everywhere is different.

Line 78 check for a possible typo «of 25»

Line 189 pore formation ≠ membrane damage?

Line 262 Ros should be changed to H2O2

Line 408 section number

Line 120, 122 Bars should be changed to stripes or smth.

It has a thick wall, as well as a capsule, which is difficult for any molecule to overcome. Thus, the possibility of interaction between peptides and the cell membrane is questionable. Do you take into account the fact of the presence of a cell wall in your reasoning?

Author Response

Authors´ Response to Reviewer #2

Reviewer #2 Comments to the Authors 1

Dear authors, there are several comments and questions to your article. This article (2) is a continuation of Antifungal Potential of Synthetic Peptides against Cryptococcus neoformans: Mechanism of Action Studies Reveal Synthetic Peptides Induce Membrane–Pore Formation, DNA degradation, and apoptosis (1). The experiments in both papers are well designed and deserve close attention. However, due to the endless references to article (1), article (2) lacks data. It is necessary to expand the Introduction section, more clearly outlining the range of issues addressed in the text.

Authors’ General Response #1

Dear Reviewer #2

We are thankful for you expending time to review our manuscript and for all your comments. We have to assume that the article is much better now after all your criticisms and suggestions. Thank you. The introduction was expanded as requested.

Reviewer #2 Comments to the Authors 1

It is necessary to characterize the studied peptides. Give the amino acid sequence and describe in detail what are the differences between the peptides, in relation to the possible mechanism of action. You should also understand the abbreviation for peptides. In the abstract, SP is used, in the introduction SAMP, hereinafter it is found simply - peptides. The choice of specific peptides should be justified: for what reasons this sequence was recognized as promising.

Authors’ General Response #1

Dear Reviewer #2

We understand your concern about the abbreviations, we fixed and choose only SAMPs.

We understand your concern about the peptides´ characterization and all information about them. Those peptides were designed by our research group back in 2016. The information your required was already published. That´s why we cannot put again in this study. The choice of specific peptides should be justified: for what reasons this sequence was recognized as promising were explained in those manuscript for each peptide. Please, see the published studies below:

RcAlb-PepII and RcAlb-PepIII à Dias et al., 2020 - https://doi.org/10.1016/j.bbamem.2019.183092  

PepGAT and PepKAA à Souza et al., 2020 - https://doi.org/10.1016/j.biochi.2020.05.016

As requested by you, the sequences of peptides were added in the M&M.

Reviewer #2 - Comment 1

Figure 1. No scale lines. Control and experiment were taken at different magnifications. The white balance is out of alignment, causing blue to appear green. The text indicates "blue dots", which are not on some pictures.

Authors’ comments #1

The correct scales were added. Indeed, control and treated were taken in different zooms. Control at 100 µm and treated at 50 µm to clearly see the colored cells. We agree with you. Some dots are more likely cyan color. We added this information in the caption and in the text.

Reviewer #2 - Comment 2

Figure 2. In the PepGat (fluorescence) figure, there is a green spot that does not correspond to any of the cells shown in the illustration without fluorescence. In other figures, elongated cells are noticeable, if this is an artifact, it must be indicated in the text.

Authors’ comments #2

Dear reviewer #2,

What happened in the figure could be explained. We first take the pictures of fluorescence field. And at the time we moved to light field there a movement of cells in the coverslip. We do these analyses in solution, them the Brownian movement change the position of the cells. If you look in the panel of PepGAT we have triple cells with green fluorescence and a small one cell alone in right and in top also with green fluorescence. If you look carefully wanna see the same pattern in the cell movement in the panel of RcAlb-PepII. The cells in the light field are slightly above of the cell in the fluorescence field, proving that the cells move in the coverslip. Yes, the elongated cells were artifact of damage caused by peptides in cells leading them to death. If you look to control, you cannot see those elongated cells.

Reviewer #2 - Comment 3

Figure 4. What method was used to obtain the data for 4A? What does "percent inhibition" mean? It would be more informative to provide data on cell concentration after exposure to peptides in each experiment (with and without ascorbic acid). This question arises since in the above images the number of cells per area unit under the influence of peptides is noticeably lower than in the control. At the same time, the text of the article says that the peptides do not exhibit antifungal activity. In the signature there is no decoding of the letters «a, b, c, d» given on 4A.

Authors’ comments#3

The data in Fig. 4A was obtained by doing the inhibitory activity of peptides in the presence and absence of AsA.

The percent inhibition in the value of inhibition of peptides in the presence and absence of AsA. This percent of inhibition is based on cell abs at 630 nm. So, it reflects the cell concentration. As lower number of cells, higher is the inhibition and vice-versa. To perform the methodology, we followed the study published by Neto et al. 2020 ( doi:10.1016/J.IJBIOMAC.2019.09.142).

The meaning of lowercase letters meaning was added.

“ The different lowercase letters in (A) indicates statistical significance at p > 0.05.”

Reviewer #2 - Comment 4

Figure 5. The same. The concentration of cells under the influence of peptides is lower than in the control, in the absence of fluorescence. It is necessary either to replace the pictures, or to explain this effect.

Authors’ comments#4

If you look carefully, the peptides have a reduction in the activity in the presence of AsA. But still maintain some activity. Only RcAlb-PepII completely lost its activity, and in the figure 5, compared to other peptides, RcAlb-PepII present the higher number of cells and is in consonance with no fluorescence detected. We have already explained and discussed that in the discussion.

Reviewer #2 - Comment 5

MIC or MIC50? Everywhere is different.

Authors’ comments#5

Thank you for this suggestion. We have used only MIC50.

Reviewer #2 - Comment 6

Line 78 check for a possible typo «of 25»

Authors’ comments#6

It was checked, thank you.

Reviewer #2 - Comment 7

Line 189 pore formation ≠ membrane damage?

Authors’ comments#7

 It was fixed, thank you.

Reviewer #2 - Comment 8

Line 262 Ros should be changed to H2O2

Authors’ comments#8

We have used ROS because we evaluated two type anion superoxide and hydrogen peroxide. Do you agree with it?

Reviewer #2 - Comment 9

Line 408 section number

Authors’ comments#9

Sorry, for that silly mistake. It was fixed.

Reviewer #2 - Comment 10

Line 120, 122 Bars should be changed to stripes or smth.

Authors’ comments#10

You right. Sorry for that silly mistake. We have fixed.

Thank you.

Reviewer #2 - Comment 11

It has a thick wall, as well as a capsule, which is difficult for any molecule to overcome. Thus, the possibility of interaction between peptides and the cell membrane is questionable. Do you take into account the fact of the presence of a cell wall in your reasoning?

Authors’ comments#11

Dear Reviewer #2

We have showed by SEM analysis that all peptides induced several damages in the C. neoformans cells Aguiar et al. [7] (doi:10.3390/pharmaceutics14081678). The analysis, revealed severe damage in the cell wall. Treated cells presented broken cell wall with many scares on it. Additionally, in the same paper [7] we discussed that all peptides interacted with chitin, which is an important component, and that interacting with the cell wall induced damage on it. Additionally, is important to tell that peptides have a very small size being the highest with barely 1 kDa.

Round 2

Reviewer 2 Report

Dear authors, I believe that the text needs some more corrections.

«The percent inhibition in the value of inhibition of peptides in the presence and absence of AsA. This percent of inhibition is based on cell abs at 630 nm. So, it reflects the cell concentration. As lower number of cells, higher is the inhibition and vice-versa. To perform the methodology, we followed the study published by Neto et al. 2020 ( doi:10.1016/J.IJBIOMAC.2019.09.142).»

The method should be described in the materials and methods with reference to the required article.

«Yes, the elongated cells were artifact of damage caused by peptides in cells leading them to death. If you look to control, you cannot see those elongated cells.»

This fact is very interesting. The presence of cell deformation caused by peptides should be mentioned in the text.

«We have used ROS because we evaluated two type anion superoxide and hydrogen peroxide. Do you agree with it?»

Then it is worth referring to several figures, since in Figure 4 the data are only for H2O2.

Signatures need to be brought into line with the text of the article. "Synthetic peptides" should be replaced by "SAMPs" or "synthetic antimicrobial peptides".

Author Response

Authors´ Response to Reviewer #2

Reviewer #2 Comment 1

Dear authors, I believe that the text needs some more corrections.

«The percent inhibition in the value of inhibition of peptides in the presence and absence of AsA. This percent of inhibition is based on cell abs at 630 nm. So, it reflects the cell concentration. As lower number of cells, higher is the inhibition and vice-versa. To perform the methodology, we followed the study published by Neto et al. 2020 ( doi:10.1016/J.IJBIOMAC.2019.09.142).»

The method should be described in the materials and methods with reference to the required article.

Authors’ Response #1

Thank you, dear reviewer #2, and sorry for the silly mistake. The information was added.

Reviewer #2 Comment #2

«Yes, the elongated cells were artifact of damage caused by peptides in cells leading them to death. If you look to control, you cannot see those elongated cells.»

This fact is very interesting. The presence of cell deformation caused by peptides should be mentioned in the text.

Authors’ General Response #1

We added the information

Reviewer #2 - Comment 3

«We have used ROS because we evaluated two type anion superoxide and hydrogen peroxide. Do you agree with it?»

Then it is worth referring to several figures, since in Figure 4 the data are only for H2O2.

Authors’ comments #4

We fixed in text. But Fig. 4 is correct. In this figure we detected only H2O2 and in Fig. 1 we detected O2•–. When we talk about ROS we are referring to both O2•– and H2O2.

Reviewer #2 - Comment 4

Signatures need to be brought into line with the text of the article. "Synthetic peptides" should be replaced by "SAMPs" or "synthetic antimicrobial peptides".

Authors’ comments #2

We fixed all across the text.